Patients with lower BMI are more likely to experience shoulder pain after single port laparoscopic myomectomy

Chen Xiubin nice2seeu78@hotmail.com 1
Guo Min 1
Pei Zheng 2
1 Department of Anesthesiology, Shanghai First Maternity and Infant Hospital, Tongji University School of Medicine , Shanghai , China
2 Department of Nursing, Shanghai First Maternity and Infant Hospital, Tongji University School of Medicine , Shanghai , China
Anson Lesley
Electronic publication date: 2025 Nov 21
Publication date: 2025
Volume: 13
Electronic Location ID: e20362
Received 2025 May 8; Accepted 2025 Oct 17
Copyright: ©2025 Chen et al.
Copyright year: 2025
Copyright holder: Chen et al.
License: This is an open access article distributed under the terms of the Creative Commons Attribution License, which permits unrestricted use, distribution, reproduction and adaptation in any medium and for any purpose provided that it is properly attributed. For attribution, the original author(s), title, publication source (PeerJ) and either DOI or URL of the article must be cited.
License URL: https://creativecommons.org/licenses/by/4.0/

Keywords: PLSP, Shoulder pain, Single-port laparoscopic surgery, Myomectomy

Funding: The authors received no funding for this work.

==============================
Objective

The aim of this study is to evaluate the incidence of shoulder pain after single port transumbilical laparoscopic myomectomy and analyze patient and operative specific factors associated with post-laparoscopy shoulder pain (PLSP). This information can inform preoperative risk assessment and intervention.

Method

This is a prospective cohort study. Two hundred and twenty patients undergoing elected single-port laparoscopic myomectomy under general anaesthesia were divided into two groups according to whether they had shoulder pain after surgery. Patient demographic data and duration of surgery, intraoperative blood loss volume, intraoperative carbon dioxide (CO2) usage volume, the usage of an abdominal drainage tube or not, the usage of patient-controlled intravenous analgesia (PCIA) or not were recorded. Statistical analyses were performed using SPSS. Data were compared using Student’s t-test or chi-square test. Risk factors were analyzed using logistic regression.

Results

The incidence of shoulder pain was 43.18%. The body mass index (BMI) of the shoulder pain group was significantly lower than that of the non-shoulder pain group (OR = 0.629, p < 0.05). Patients with BMI < 21.64 are more likely to experience postoperative shoulder pain. The use rate of abdominal drainage tubes was higher in the non-shoulder pain group (OR = 0.509, p < 0.05).

Conclusion

PLSP is more likely to occur in patients with lower BMI (<21.64), and placing an abdominal drainage tube can reduce the occurrence of shoulder pain.

Introduction

Compared to open surgery, a laparoscopy approach has the advantage of small incisions, which improves cosmesis, as well as postoperative recovery and pain. For abdominal surgery, a single port transumbilical laparoscopic approach achieves satisfactory ‘scar-free’ cosmetic effects, which is welcomed by patients. However, this laparoscopy approach has been associated with an incidence of post-laparoscopy shoulder pain (PLSP). The incidence of PLSP reported in different studies ranges from 30% to 90% depending on surgical methods (Park, 2020). The cause of PLSP has not been fully elucidated and is believed to be caused by multiple factors, and is associated with referred pain (Radosa et al., 2019). There have been many studies on the prevention and treatment of PLSP, but there is relatively little research on the influencing factors of its occurrence. Our study aims to identify the occurrence of PLSP after single-port transumbilical laparoscopic myomectomy and to evaluate the influence of factors associated with PLSP. This will inform pre-operative screening to identify patients at risk for PLSP and to inform intervention.

Method

This was a prospective, observational cohort study, entered in the Chinese Clinical Trial Registry (ChiCTR2300071170). Ethics clearance was obtained from the Shanghai First Maternity and Infant Hospital Research Ethics Board (2023 ethics review No. 78). The study began on May 8, 2023, until sufficient cases were included. Patients were recruited from the west campus of our hospital. All enrolled subjects provided written informed consent.

Inclusion criteria: patients undergoing elected single-port laparoscopic myomectomy under general anesthesia, ASA physical status I or II, height ≥ 150 cm, age ≥ 18 years, capable of comprehending the study methodology.

Exclusion criteria: emergency surgery, conversion to open surgery, shoulder disease (history of fractures, shoulder periarthritis, chronic shoulder pain, etc.), long-term use of painkillers, cognitive impairment, and unable to comprehend the study methodology, or unwilling to participate in this study.

During the preoperative assessment, baseline patient characteristics, including age, height, weight, and medical history, were documented, and body mass index (BMI) was calculated.

All patients underwent comparable general anesthesia protocols. No preoperative sedation was administered. Upon arrival in the operating room, continuous monitoring of pulse oxygen saturation, blood pressure, and heart rate was initiated. General anesthesia was performed with midazolam (0.5 mg/kg), sufentanil (0.4 μg/kg), etomidate (0.3 mg/kg), and cisatracurium (0.15 mg/kg). Anesthesia was maintained with sevoflurane, propofol, and remifentanil. Cisatracurium was used to obtain intraoperative muscle relaxation. All patients received mechanical ventilation using volume-controlled mode with a tidal volume of eight ml/kg. The respiratory rate was titrated to maintain end-tidal CO2 (PetCO2) between 35–45 mmHg.

All patients were positioned in Trendelenburg during the procedure. Laparoscopic surgery was performed using CO2 pneumoperitoneum maintained below 15 mmHg. Transumbilical single-port laparoscopic techniques were employed by experienced surgeons. Upon completion, CO2 was evacuated through manual abdominal compression with an open trocar. After the surgery, patients can choose patient-controlled intravenous analgesia (PCIA) on their own. We monitored all patients in the Post-Anesthetic Care Unit (PACU) until they met stability criteria for transfer to their regular wards. The following outcomes were collected after surgery for all participants: duration of surgery, intraoperative volume of blood loss, intraoperative CO2 volume used, placement or not of an abdominal drainage tube, and use or not of PCIA. Incision pain and PLSP were assessed on the day of surgery (postoperative day (POD) 1 (POD1) and POD2. The Visual Analog Scale (VAS) is used to quantify pain intensity, with 0 indicating no pain, 1–3 indicating mild pain, 4–7 indicating moderate pain, and 8–10 indicating severe pain. If patients have shoulder pain, they are also required to describe the characteristics of the pain, whether it is left, right, or bilateral, whether it is persistent or paroxysmal. Then, patients were stratified into two groups based on the presence or absence of postoperative shoulder pain.

Data analysis was performed using SPSS 21.0 (IBM Corp)

Sample size calculation: According to other studies, the incidence of postoperative shoulder pain is about 60%, and logistic regression is proposed to screen for risk factors. Using the EPV (events per variable) method with 7 covariates (age, BMI, duration of surgery, blood loss volume, CO2 volume, usage of drainage tube, usage of PCIA), a sample size of 70 was calculated. To compare the differences between two groups using a t-test, take α = 0.05, β = 0.1, the calculated sample size is 182. Taking the larger value of the two. Considering that there may be 20% missing data, the final calculation included 220 cases.

Continuous variables were presented as median ± standard deviation and compared between groups using an unpaired Student’s t-test. Categorical variables were analyzed using either the chi-square test or Fisher’s exact test, as appropriate. Multivariate logistic regression was employed to identify significant risk factors for PLSP.

To evaluate the predictive performance of BMI for PLSP, a receiver operating characteristic (ROC) curve analysis was planned. BMI was used as the test variable, and the presence of PLSP was used as the state variable. The optimal cutoff value was determined by identifying the point on the ROC curve that maximized Youden’s index (J = sensitivity + specificity −1).

Result

Between May 8 and November 10, 2023, 223 patients who underwent elective single-port gynecological myomectomy who met the inclusion criteria; three were excluded due to shoulder disease. We divided the 220 patients into two groups according to whether they had shoulder pain after surgery: 95 (43.18%) who had shoulder pain as the SP group, and the other 125 (56.82%) who didn’t have shoulder pain as the NSP group.

Shoulder pain in this cohort was predominantly right-sided or bilateral; isolated left-sided pain was far less frequent. The majority of cases (n = 59, 62.1%) reported shoulder pain on the first day after surgery, following initial ambulation. The majority of patients presented with mild to moderate pain, while a subset (10%) experienced severe pain. The pain was usually described as dull, aching, or distending pain, and most was paroxysmal pain, occurred or worsened when getting up for activities or changing body positions. Some can be alleviated by massaging. Table 1 summarizes the baseline characteristics of patients experiencing postoperative shoulder pain.

Table 1 Incidence and characteristics of PLSP.

NSP	125 (220)	56.82%	SP	95 (220)	43.18%	
			N (95)	% (100%)		
Site of shoulder						
Bilateral shoulder			40	42.11%		
Right shoulder only			49	51.58%		
Left shoulder only			6	6.32%		
Presence of shoulder pain						
On the day of surgery			34	35.79%		
On the first day after surgery			59	62.11%		
On the second day after surgery			2	2.11%		
Degree of shoulder pain						
Moderate pain			56	58.95%		
Mild pain			19	20%		
Severe pain			10	10.53%		
Pain pattern						
Paroxysmal			92	96.84%		
Persistent			3	3.16%		
Notes.

Out of 220 patients, 95 (43.18%) had shoulder pain as the SP group, 125 had no shoulder pain as the NSP group. Values are presented as number and proportion.

There were no significant differences between the two groups in terms of age, duration of surgery, intraoperative blood loss volume, intraoperative carbon dioxide (CO2) usage volume, and the usage of PCIA. The BMI of group NSP is much smaller than group SP (p < 0.05). The usage rate of an abdominal drainage tube in group NSP is higher than that in group SP (p < 0.05). The comparison of the two groups is shown in Table 2.

Table 2 Comparison of demographic and perioperative events between the SP and NSP groups.

	Group SP	Group NSP	p-value	
Age (year)	37.99 ± 5.46	39.13 ± 5.56	0.131	
BMI (kg/m2)	20.93 ± 2.29	24.09 ± 3.18	0.000*	
Duration of surgery (min)	94.53 ± 35.67	103.82 ± 42.86	0.088	
Blood loss volume (ml)	63.47 ± 74.42	62 ± 74.12	0.884	
CO2 volume (L)	235.26 ± 194.86	247.37 ± 173.72	0.628	
Usage of PCIA	87 (91.6%)	118 (94.4%)	0.431	
Usage of drainage tube	47 (49.5%)	82 (65.6%)	0.019*	
Notes.

Data are presented as the mean ± SD. Group SP: shoulder pain, Group NSP: no shoulder pain.

* p < 0.05.

Univariate binary logistic regression analysis identified both BMI (OR = 0.629, p < 0.05) and the usage of a drainage tube (OR = 0.509, p < 0.05) as significant protective factors against post-laparoscopic shoulder pain (Table 3). The data reveal a significant inverse association between BMI and PLSP, with lower BMI values associating with increased PLSP incidence (Table 4).

Table 3 Univariate binary logistic regression analysis.

	OR	95% CI	p-value	
Age (year)	0.994	0.937–1.054	0.833	
BMI(kg/m2)	0.637	0.553–0.733	0.000*	
Duration of surgery (min)	0.999	0.988–1.010	0.830	
Blood loss volume (ml)	1.003	0.998–1.008	0.215	
CO2 volume (L)	1.000	0.998–1.003	0.829	
Usage of drainage tube	0.505	0.259–0.983	0.044*	
Usage of PCIA	0.282	0.070–1.142	0.076	
Notes.

* p < 0.05.

Table 4 The incidence of shoulder pain in patients with different BMI (kg/m2) cutoff values.

	Shoulder pain	No shoulder pain	Chi square	p-value	
BMI < 18.5	13 (81.3%)	3 (18.8%)	30.247	0.000*	
18.5 ≤ BMI < 24	69 (50.4%)	68 (49.6%)	
24 ≤ BMI < 28	13 (24.1%)	41 (75.9%)	
BMI ≥ 28	0 (0%)	13 (100%)	
Notes.

* p < 0.05.

To evaluate the predictive value of BMI values for PLSP, we used different BMI values as the test variable and no shoulder pain as the state variable, and plotted a receiver operating characteristic (ROC) curve with an area under the curve of 0.788 (Fig. 1). The optimal BMI threshold was determined by identifying the point on the ROC curve that maximized the Youden’s index (J = sensitivity + specificity −1). A BMI value of 21.64 kg/m2 was identified as the optimal cutoff for predicting the absence of PLSP. At this threshold, the sensitivity was 0.792, the specificity was 0.663, the positive likelihood ratio was 2.351, and the negative likelihood ratio was 0.314.

Figure 1 ROC curve made using BMI(kg/m2) values and no shoulder pain.

Discussion

Single-port laparoscopic surgery is increasingly widely used in gynecological surgeries due to its advantages of fewer incisions, less pain, and a “scar-free” cosmetic effect. However, PLSP, a unique complication of laparoscopic surgery, often troubles the patients. The incidence of postoperative laparoscopic shoulder pain (PLSP) in our study was 43.18%, consistent with the documented prevalence rates of 30–90% in previous studies.

The cause of PLSP has not been fully elucidated and is believed to be caused by multiple factors, and is associated with referred pain. The phrenic nerve is mainly composed of the anterior branch of the C4 spinal nerve root, and its sensory fibers are distributed in the diaphragm, mediastinal pleura, pericardium, and part of the peritoneum below the diaphragm. The C4 nerve also provides sensory nerve innervation for the shoulder skin. Therefore, pain caused by stimulation of the diaphragm or damage to the phrenic nerve traction will be transmitted to the shoulder through the phrenic nerve (Kim et al., 2021). There are at least three possible theories to explain the cause of PLSP. Carbonic acid theory: The production of carbonic acid lowers peritoneal pH, irritating the peritoneal and diaphragmatic nerves (Kaloo et al., 2019). Residual gas theory: Retained CO2 in the abdominal cavity mechanically stimulates nerves (Sao et al., 2019). Tissue trauma theory: Pneumoperitoneum-induced stretching or damage to the peritoneum and diaphragm leads to vascular tears, phrenic nerve traction, and inflammatory mediator release, resulting in referred shoulder pain (Donatsky, Bjerrum & Gögenur, 2013).

Our research suggests that patients with lower BMI are more likely to have PLSP. This may be related to the large upper abdomen space of thin patients in the Trendelenburg position, which may accumulate a large amount of gas after operation (Li et al., 2021). Clinical studies have demonstrated that residual intraperitoneal gas may exist following laparoscopic procedures, with a significantly higher incidence observed in patients with low BMI (Gayer et al., 2000). And there is a positive correlation between residual pneumoperitoneum and shoulder pain intensity after laparoscopic surgery (Song, Kim & Lee, 2017). Our finding is consistent with the positive correlation previously reported between residual pneumoperitoneum and shoulder pain intensity after laparoscopy.

There are many strategies to prevent and mitigate the severity of PLSP. One method is to use low-pressure pneumoperitoneum. While the standard insufflation pressure for gynecologic laparoscopy typically ranges from 12–14 mmHg (with 15 mmHg as the upper limit), this pressure range itself contributes to adverse effects, including PLSP (Neudecker et al., 2002). Clinical investigations revealed significantly lower PLSP intensity in patients who received laparoscopic procedures performed at 7–8 mmHg pneumoperitoneum pressure (Radosa et al., 2019; Sroussi et al., 2017). However, due to the inherent technical difficulties, such as interference between instruments during operation, limitations in the visual field, and missing surgical triangles, the difficulty of single-port laparoscopic surgery increases. Using low pressure during pneumoperitoneum may lead to insufficient abdominal insufflation, resulting in a suboptimal surgical field. This can hinder the clear identification of anatomical structures and target lesions, potentially increasing the risk of intraoperative complications. Therefore, due to the uncertain safety of using low-pressure pneumoperitoneum, it is not suitable for single-port laparoscopic surgery. However, recent research shows that deep neuromuscular blockade and low-pressure pneumoperitoneum can provide better surgical conditions (Özdemir-van Brunschot et al., 2018), but further research is still needed on its safety and whether it can reduce PLSP.

Given that CO2 pneumoperitoneum represents a primary etiological factor for PLSP, it may be beneficial to expel as much gas as possible after laparoscopic surgery to reduce the amount of residual gas in the abdominal cavity. In our study, CO2 was evacuated through manual abdominal compression to reduce gas residue in the abdominal cavity. However, the incidence of PLSP is still 43.18%. Researches demonstrate that pulmonary recruitment maneuvers (PRM) significantly reduce both residual pneumoperitoneum volume and post-laparoscopic shoulder pain incidence (Garteiz-Martínez et al. 2021; Lee et al., 2020; Kietpeerakool et al., 2020). At laparoscopic surgery termination, PRM is performed by: maintaining Trendelenburg positioning, administering five positive-pressure breaths (30–60 cm H2O) with a 5-second hold on the final breath, and simultaneously applying mild abdominal compression. This technique helps expel CO2 through open trocar sites. The underlying principle is to expand the lungs via positive-pressure ventilation, elevate intrathoracic pressure, lower the diaphragm, increase intra-abdominal pressure, and facilitate CO2 elimination from the peritoneal cavity, thereby minimizing residual gas. Future studies may employ this technique to verify its effectiveness in single-port laparoscopic surgery.

Another clinical method is available for reducing postoperative pneumoperitoneum after laparoscopic surgery. Our research shows that the usage of a drainage tube is a protective factor against shoulder pain after single-port laparoscopy (OR = 0.509). However, the efficacy of abdominal drainage tube placement in reducing post-laparoscopic shoulder pain (PLSP) remains controversial, with inconsistent findings in the current literature (Nursal et al., 2003; Craciunas, Stirbu & Tsampras, 2014).

Given the limited efficacy of analgesics in managing PLSP, greater emphasis should be placed on preventive strategies. Our study demonstrates that patients with a lower BMI are at increased risk of PLSP, particularly those with a BMI below 21.64, who warrant special clinical attention in future practice. According to our findings, intraperitoneal drainage appears to be an effective intervention. Additional reported strategies for PLSP mitigation include: PRM, utilization of warmed and humidified insufflation gas, and intraperitoneal fluid administration, etc. Future research should focus on formally evaluating the efficacy of these economical and low-tech interventions in high-risk PLSP populations, with all investigations conducted under the primary condition of ensuring patient safety.

Our study has several limitations that should be considered when interpreting the findings. Firstly, its observational design inherently precludes the establishment of causality, and residual confounding from unmeasured factors may influence the observed associations. Secondly, the generalizability of the results is limited by the exclusive inclusion of female patients, leaving the applicability to male populations uncertain. Thirdly, the relatively small sample size may have compromised statistical power and affected the stability of the estimates. Despite the limitations mentioned above, our study still provides valuable foundational evidence for the association between PLSP and BMI, especially for the gynecological patient population. The research results not only reveal the important correlation between the two but also provide direction and basis for future prospective studies on a larger scale, including mixed populations, to verify and expand the observations of this study.

Supplemental Information

Supplemental Information 1 Raw data

We extend our gratitude to gynecologists Siji Lv and Xiaohui Hu for their invaluable assistance in patient management during our study.

Additional Information and Declarations

Competing Interests

Author Contributions

Human Ethics

Clinical Trial Ethics

Data Availability

Clinical Trial Registration

The authors declare there are no competing interests.

Xiubin Chen conceived and designed the experiments, performed the experiments, analyzed the data, prepared figures and/or tables, authored or reviewed drafts of the article, and approved the final draft.

Min Guo performed the experiments, analyzed the data, prepared figures and/or tables, and approved the final draft.

Zheng Pei performed the experiments, authored or reviewed drafts of the article, and approved the final draft.

The following information was supplied relating to ethical approvals (i.e., approving body and any reference numbers):

Ethics clearance was obtained from the Shanghai First Maternity and Infant Hospital Research Ethics Board (2023 ethic review No. 78).

The following information was supplied relating to ethical approvals (i.e., approving body and any reference numbers):

We conducted an observational cohort study and registered at Chinese Clinical Trial Registry (ChiCTR2300071170).

The following information was supplied regarding data availability:

The raw measurements are available in the Supplemental File.

The following information was supplied regarding Clinical Trial registration:

ChiCTR2300071170.

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
