# Peer review of "Patients with lower BMI are more likely to experience shoulder pain after single port laparoscopic myomectomy"

_PeerJ, doi:10.7717/peerj.20362_

## Round 0.1 · original submission · Minor Revisions

· Academic Editor

Minor Revisions

Please correct the typo in Figure 1 ("inraorl" -> "intraoral") and check the manuscript and figures/tables carefully before resubmitting.

·

Basic reporting

Interesting retrospective study on the risk factors related with postoperative shoulder pain after single port myomectomy.
The authors assessed patients BMI, age, duration of surgery, intraoperative bleeding volume, intraoperative CO2 usage volume and the usage of PCIA.
The authors conclude that patients with lower BMI are more likely to have PLSP.

The Title could be improved. I would suggest patients instead of individuals or just: 'shoulder pain after single port laparoscopic myomectomy' or Risk factors of shoulder pain after single port laparoscopic myomectomy

The authors conclude that placing 'an abdominal drainage tube can reduce the occurrence of shoulder pain'. The authors correctly mention that this is not supported by published research (Craciunas et al) and therefore, the conclusion should be changed. Ther use of peritoneal drain could be mentioned but it is not recommended.

Experimental design

as above

Validity of the findings

as above

Additional comments

as above

Reviewer 2 ·

Basic reporting

English grammar will need to be improved throughout the text. As well, first-person language needs to be used throughout.

The structure of the Introduction needs revision to be impactful. Specifically, the introduction needs to include a summary of key findings from previous studies regarding PLSP (with supporting references). There should be clear statement on what is known and remaining gap on this issue. A hypothesis on the plausible mechanism of PLSP should be provided, leading to the statement of research aim.

Supporting references are needed in the Introduction.

The Methods are generally well-structured. I have provided specific feedback in the pdf to improve the clarity of the text, the logical organization of the information presented, and the statistical analysis.

The results need to be revised to avoid duplication of information presented in the Tables and in the text. The text should highlight the key findings of the study. Do note that that the ROC analysis was not described in the Methods section. It would be important to clearly state the BMI cutoffs used in your ROC analysis.

The Discussion is well-structured. Again, specific comments are provided in the pdf. What will need to be included is a statement of the limitations of the study.

The Conclusions are appropriate.

Experimental design

The research question is well-defined and the significance is described. As previously stated, the specific gap that the study addresses needs to clearly articulated in the Introduction.

The methods are appropriate. It is not clear why the study was registered as clinical trial.

The statistical analysis does need to be described in greater detail (i.e., including the ROC analysis) and corrected (unpaired Student's t-test). As well, terminology has to be specific (i.e., regression is not a correlation).

Validity of the findings

The findings are clinically relevant and are consistent with the research design and the planned statistical analysis.

Annotated reviews are not available for download in order to protect the identity of reviewers who chose to remain anonymous.

---

## Round 0.2 · Minor Revisions

· Academic Editor

Minor Revisions

Thank you for submitting a revised manuscript in response to our reviewers' comments. We note, however, that you have not responded to, or fully addressed, all of reviewer 2's concerns. We therefore require submission of a further revision that includes appropriate changes that address all the reviewers' concerns. The resubmission should be accompanied by a point-by-point response that quotes the entirety of the reviewers' reports from the previous round of review, responds to every comment in turn, and explains how you have revised your manuscript to address each concern.

---

## Round 0.3 · Minor Revisions

· Academic Editor

Minor Revisions

Thank you for submitting a revised manuscript and response to our reviewers' comments. Unfortunately, there remain comments in reviewer 2's report that you have not quoted or responded to (for example, those about English grammar, duplication, limitations, registration, and statistical terms). Please therefore submit a further revision that includes appropriate changes that address all of reviewer 2's concerns. Please also attach a point-by-point response that quotes the whole of reviewer 2's report from the first round of review, responds to every comment in turn, and explains how you have revised your manuscript to address each concern. Please note that we would be unlikely to offer you more than this one last chance to adequately address this reviewer's concerns.

---

## Round 0.4 · accepted · Accept

· Academic Editor

Accept

Thank you for the further revisions you have made to your manuscript and for the full response to our reviewers' comments. I am now satisfied that our reviewers concerns have been addressed and that the manuscript is ready for publication.